# The Role of the MCM2-7 Helicase Subunit MCM2 in Epigenetic Inheritance

**DOI:** 10.3390/biology13080572

**Published:** 2024-07-29

**Authors:** Jing Jia, Chuanhe Yu

**Affiliations:** Hormel Institute, University of Minnesota, Austin, MN 55912, USA; jia00165@umn.edu

**Keywords:** MCM2, Dpb3, Dpb4, Pol1, Pol32, eSPAN, FACS, cell differentiation, histone chaperone

## Abstract

**Simple Summary:**

The MCM2-7 helicase plays a crucial role in unwinding the double-stranded DNA template during DNA replication. Recent studies indicate that the MCM2-7 helicase subunit Mcm2 also facilitates the transfer of the parental histone H3-H4 tetramer (parH3:H4tet) onto the newly replicated lagging strand. This review highlights recent advances in our understanding of the role of the replication machinery, particularly Mcm2, in epigenetic inheritance. Disruption of this function can affect chromatin stability, DNA repair processes, and cell differentiation.

**Abstract:**

Recycling histone proteins from parental chromatin, a process known as parental histone transfer, is an important component in chromosome replication and is essential for epigenetic inheritance. We review recent advances in our understanding of the recycling mechanism of parental histone H3-H4 tetramers (parH3:H4tet), emphasizing the pivotal role of the DNA replisome. In particular, we highlight the function of the MCM2-7 helicase subunit Mcm2 as a histone H3-H4 tetramer chaperone. Disruption of this histone chaperone’s functions affects mouse embryonic stem cell differentiation and can lead to embryonic lethality in mice, underscoring the crucial role of the replisome in maintaining epigenomic stability.

## 1. Introduction

DNA replication is a fundamental process for transmitting genetic information from parent to daughter cells. Epigenetics is defined as heritable changes in gene expression that occur without alterations to the underlying genomic sequence [1]. Epigenetic inheritance is vital for the establishment and maintenance of cell lineages, significantly influencing cell fate determination. 

Eukaryotic genomic DNA is organized into a chromatin structure. The fundamental unit of chromatin, the nucleosome, includes two copies each of histones H2A, H2B, H3, and H4, which together form an octameric core. Around this core, 147 base pairs of DNA are tightly wound [2]. During DNA replication, the replication fork disrupts the core histone octamer. After the replication fork passes, the octamer is reassembled with a combination of parental and newly synthesized histones [3,4]. The recycling of parental histones H2A/H2B and H3/H4 occurs differently during DNA replication. Radioisotope labeling experiments demonstrate that histone H3-H4 tetramers are transferred intact to the newly synthesized DNA without segregating [5]. In contrast, histones H2A-H2B dimers dissociate from the parental octamer and randomly reincorporate into newly assembled nucleosomes [6,7]. In recent studies, the histone H2A-H2B dimer was found to segregate symmetrically to the two daughter strands [8]. Furthermore, the structural analysis implied that the parental H2A-H2B dimers could be recycled as a form of hexamer with parental H3-H4 tetramer (parH3:H4tet) [9]. The transfer of parH3:H4tet during chromatin replication is the first and most important step in epigenetic inheritance [10,11]. Following the recycling of parental histones, the newly synthesized DNA daughter strands also recruit newly synthesized histones to maintain consistent nucleosome structures. The faithful transfer and restoration of post-translational histone modifications reflective of parental histone states are critical for maintaining cell identity throughout genome duplication.

Four major H3-H4 chaperones—CAF1 (chromatin assembly factor 1), Asf1 (anti-silencing factor 1), Rtt106 (regulator of Ty1 transposition factor 106), and FACT—have been identified as key participants in the nucleosome assembly of newly synthesized H3-H4 tetramers (nsyH3:H4tet) in budding yeast. The H3-H4 histone chaperone CAF1 is highly conserved from yeast to human cells and mediates nucleosome assembly in a DNA replication-dependent interaction with proliferating cell nuclear antigen (PCNA) [10,12]. Mutations in CAF1 affect transcriptional silencing from yeast to mammals [13,14,15]. The chaperone Asf1 binds nsyH3:H4tet and this binding is required for acetylation of H3K56. H3K56Ac is important for nucleosome assembly following DNA replication and DNA repair [11,16]. Although the Asf1-bound H3-H4 is proposed to be transferred to CAF1 or Rtt106, Asf1 and CAF1 are only partially redundant, as double mutant cells show greater gene silencing defects than Asf1 mutants alone [17]. The chaperone Rtt106, unique to budding yeast, exacerbates the silencing defects of CAF1 but not Asf1 mutations; this function is dependent on Rtt106’s role as a histone chaperone [18,19,20]. In addition, replication protein A, a single strand DNA binding protein, is also involved in nsyH3:H4tet deposition and assembly in a replication and transcription dependent manner [21,22]. Despite extensive studies on nsyH3:H4tet assembly, many questions remain about the transfer or recycling of parH3:H4tet. For instance, how are parH3:H4tet transferred to the replicating chromatin? Are old and newly synthesized histones assembled into nucleosomes at the leading and lagging DNA strands differently? How are histone markers, such as methylation and acetylation, copied from the parental histones to newly synthesized histones following DNA replication to maintain epigenetic information? These fundamental questions, related to parental histone transfer, are highly significant as they are related to human disease pathology and stem differentiation [18,23,24,25]. In recent years, advances in technology have enabled the exploration of mechanisms underlying the recycling of parental histone H3-H4. In this manuscript, we will focus on the role of Mcm2 in this process.

## 2. The eSPAN Method to Study DNA Replication-Coupled Epigenetic Inheritance

The molecular mechanisms of DNA replication have been studied using various techniques, including gradient centrifugation, two-dimensional gel electrophoresis, electron microscopy, chromatin immunoprecipitation, fluorescence microscopy, and DNA fiber analysis [26,27,28]. Detecting the replication strand preference of histones is imperative to addressing the questions noted above. However, determining whether a protein is located on the leading or lagging strand of replicating DNA using standard biochemical methods is challenging. We developed a new method called eSPAN (enrichment and sequencing of protein-associated nascent DNA) [29] to allow us to distinguish proteins bound to the leading and lagging strands of newly synthesized DNA (Figure 1). In eSPAN, nascent DNA is first labeled with BrdU or EdU (5-ethynyl-2′-deoxyuridine). A replication-associated protein or modification is then subjected to ChIP. The protein-associated new DNA is isolated by BrdU IP and sequenced using a strand-distinguishing library preparation method. Using eSPAN, we have confirmed the strand preference of several key proteins, including Pol δ, Pol ε, DNA ligase, and MCM2-7 helicase. We have also found that the DNA replication clamp PCNA prefers the lagging strand under normal conditions but is specifically unloaded from the lagging strand in response to replication stress. Thus, eSPAN enables, for the first time, rigorous genome-wide analysis of replication-coupled deposition of preexisting nucleosomes onto leading and lagging strands [29,30]. Additionally, with eSPAN or a similar method known as SCAR-seq (sister chromatids after replication by DNA sequencing), several key regulators in epigenetic inheritance, including Mcm2, have been identified [31,32,33]. 

## 3. The Function of Mcm2 and Its Interaction Partners in DNA Replication

The replisome, a complex dedicated to DNA replication, comprises multiple proteins, including the CMG helicase (Cdc45-MCM2-7-GINS) and three DNA polymerases (alpha, delta, epsilon) at its core [34]. The structure of the replisome has been elucidated with cryo-electron microscopy, revealing that the MCM2-7 complex is the core structural and functional component of active replication forks. During G1, the MCM2-7 complex is recruited to replication origins by the origin recognition complex (ORC) and Cdc6/Cdt1 via their C-terminal regions, forming a head-to-head MCM2-7 double hexamer around DNA [35,36]. This assembly, including ORC, Cdc6/Cdt1 and MCM2-7 was called the pre-replicative complex (Pre-RC) [37,38]. The initiation of DNA synthesis during S phase requires the splitting and activation of the MCM2-7 double hexamer in the Pre-RC to form two copies of the CMG replicative helicase [39,40,41], that move in opposite directions, unwinding double stranded DNA ahead of replication [42].

MCM2, also known as CDCL1 and BM28, is a nuclear protein first discovered in *Saccharomyces cerevisiae* during the screening of mutants defective in minichromosome maintenance [43]. It took many years to identify its function as a replicative helicase. Besides replicative helicase, Mcm2 has also been reported to play roles in various cellular processes (such as cohesion regulation, transcription, S-checkpoint) across species, including human tumor pathology [44]. In *S. cerevisiae*, the MCM2 gene is located on chromosome II and encodes a protein with a calculated molecular weight of 99 kDa, although it appears as 120 kDa on SDS-PAGE gels [45]. The human MCM2 gene is located on chromosome 3q21 [46] and encodes a 904 amino acid polypeptide with a calculated mass of 101 kDa, observed as 125 kDa on SDS-PAGE gels [45]. The MCM2 gene is evolutionarily conserved from yeast to humans [45,47,48].

In the Mcm2 protein, the N-terminal region, the central region, and the C-terminal region are all conserved [49]. The largest and most conserved is the central region, which is approximately 200 amino acids long and contains residues present in the A motif of the Walker-type nucleoside triphosphate-binding sequence GXXGXGKS. This region is responsible for the catalytic activity of Mcm2. The N-terminal region contains a zinc finger motif (_342_CX_2_CX_19_CX_2_C_368_), which plays a critical role in MCM2 function, involving efficient ssDNA binding and helicase activity [50]. The N-terminal tail of Mcm2 is highly acidic (30% aspartate and glutamate) and conserved across all eukaryotic Mcm2 orthologs [45,48]. Additionally, the N-tail of Mcm2 contains a conserved histone-binding domain (HBD), located within amino acids 69-138 in the human Mcm2 protein [51]. This domain wraps around the lateral surface of an H3-H4 tetramer. Mutations of these conserved tyrosine residues to alanine (mcm2-3A: Y79A, Y82A, and Y91A in yeast; Y81A, Y89A, and Y137A in humans) or (mcm2-2A: Y79A, Y82A in yeast; Y81A, Y89A in humans) abolish Mcm2 binding to the H3-H4 tetramer [48].

## 4. Mcm2’s Role in Parental Histone Recycling

The HBD of Mcm2 is crucial for interaction between H3-H4 and Mcm2 and is proposed to affect the recycling of parental histones. However, mutations in the HBD of Mcm2 (mcm2-2A or mcm2-3A Table 1) only lead to a slightly higher loss of heterochromatin silencing at sub-telomeres and mating type loci [48]. Using the eSPAN method, we and others demonstrated that parental histones in mcm2-3A mutants exhibit a strong leading strand bias, contrasting with the slight lagging strand preference observed in wild-type yeast cells [31,33,52]. This parental histone transfer mechanism is conserved, as the HBD mutation of Mcm2 in mouse embryonic stem cells also shows a leading strand bias, differing from the nearly no strand bias observed in wild-type cells [32].

Until now, multiple proteins and factors in addition to Mcm2 have been identified as playing roles in the transfer of parH3:H4tet to the replication lagging strand (Figure 2) [9,31,32,53]. Ctf4 structurally forms a trimer that associates with DNA replication forks on both the leading and lagging strands of the fork [54,55]. On the leading strand, Ctf4 binds to the CMG helicase; on the lagging strand, it binds with Pol1 or other untested proteins. In ctf4-4E mutant cells, the Ctf4 protein cannot bind to Pol1. The parH3:H4tet in ctf4-4E mutants shows a strong leading-strand bias, while the nsyH3:H4tet exhibits a lagging-strand bias. Additionally, the amount of parH3:H4tet of ctf4-4E is significantly decreased on the lagging strand, which is consistent with observations in mcm2-3A cells. Similarly, in pol1-4A mutant cells, which are defective in Ctf4 binding, parH3:H4tet also shows a significant bias towards the leading strand. However, nsyH3:H4tet shows no bias towards the lagging strand in pol1-4A mutant cells [31]. Mcm2, Ctf4, and Pol1 appear to function in the same pathway for transferring parental histones to the replication lagging strand. Mutations in any of these components compromise the slight inheritance defect of the silent state at the HML locus [31].

Recently, the DNA polymerase δ subunit Pol32 was identified as being involved in the transfer of parental histones to the lagging strand (Figure 2), functioning downstream of Mcm2 [56,57]. Pol32 can bind directly with Mcm2, and their interaction is independent of Ctf4 [56]. Since the MCM2-7 helicase is located at the front of the replisome, Mcm2 is likely the first and key parental histone chaperone. Supporting this notion, parH3:H4tet, once disassembled from the parental chromatin bound to Mcm2, is then transferred from Mcm2 to Pol32, and subsequently via Pol1 to the lagging strand [9,56]. While Ctf4 supports Pol32 in the transfer of parental histones, further experiments are needed to elucidate the detailed pathways of this transfer process [56].

In contrast to the lagging strand pathway, the Pol ε subunits (Dpb3 and Dpb4) (Figure 2) are responsible for transfer of parental histones to the leading strand, a process originally discovered in budding yeast cells [33]. A later study using mouse EC cells confirmed that this leading strand parental histone transfer pathway is conserved across different species [58]. To date, there is no direct evidence for the presence of other factors in the parental histone transfer to the leading strand. Previous reports suggest that chromatin assembly on the lagging strand is more critical for chromatin epigenetic maintenance [59]. The identification of more parental histone chaperones or factors that regulate parental histone transfer to the lagging rather than leading strands supports this notion. Based on current knowledge, the transfer of parH3:H4tet to the lagging strand requires more proteins than transfer to the leading strand, implying this pathway is more complicated and therefore may be more sensitive to loss of function by mutation or deletion. The exact pathways of parH3:H4tet to the lagging strand is still unclear; the Mcm2 protein plays an important role in this pathway, mutations in which have more severe impacts on heterochromatin inheritance than dpb3∆.

## 5. Additional Mcm2 Interaction Factors in Parental Histone Transfer

During DNA replication, the CMG complex translocates along the leading strand to unwind DNA [60]. It uses the N-terminal of the MCM2-7 ring as the leading edge to interface with parental nucleosomes. Multiple accessory factors, including Ctf4, Mrc1, FACT, and Tof1-Csm3, exert regulatory roles in DNA replication, cell cycle checkpoints, and protection against replication stress. Many of these factors are involved in regulating the recycling of parental histones.

The FACT (facilitates chromatin transcription) complex, initially identified as part of the replisome and transcriptional regulation complex, comprises Spt16 and Pob3 (homologue of SSRP1 in mammals) in yeast [61]. During RNA transcription or DNA replication, FACT promotes nucleosome disassembly and reassembly, facilitating global accessibility of nucleosomes without ATP hydrolysis [62,63]. In addition to binding with free histone H2A-H2B, FACT also interacts with the H3-H4 tetramer and participates in parental histone recycling. The interaction between FACT and H3-H4 is intricate, involving direct physical binding between Spt16 and H3-H4, Mcm2-mediated interaction, and H2A-H2B-mediated interaction. During DNA replication, Mcm2 binds with Spt16 only when parental histone complexes are released from chromatin, indicating that Mcm2 protein picks up H3-H4 histones released from chromatin with the cooperation of FACT, and direct binding between Mcm2 and FACT is unnecessary [48]. Although the N-terminal of Spt16 does not directly bind with histone H3-H4, it mediates the binding between FACT and the MCM2-7 complex, thereby influencing the transfer of parH3:H4tet to the lagging strand [64]. 

Recent structural analyses propose a model in which the (H3-H4)2-(H2A-H2B) hexametric complex can be transferred as a unit during parental histone recycling, suggesting the involvement of H2A-H2B in FACT-H3-H4 interaction [9]. Based on eSPAN analyses, parental histone transfer does not show a clear strand bias in Spt16 depletion cells, but deletion of the Spt16 N-terminal region shows a slight bias toward the leading strand [64]. Overall, the FACT complex is proposed to participate in parental histone transfer on both strands; it acts in concert with Dpb3/4 for parental histone transfer on the leading strand, while transferring parH3:H4tet to the lagging strand together with the Mcm2 pathway or other unknown proteins [64].

Tof1p (topoisomerase I interacting factor) is a checkpoint-mediator protein that regulates DNA damage responses during the S phase [65]. Together with Mrc1p and Csm3p, Tof1p forms a replication-pausing mediator complex that associates with DNA replication forks [66]. Recent investigations into Tof1’s function in the parental histone recycling process combined cryo-electron microscopy and eSPAN analysis. Structurally, Tof1 interacts with the N-terminal extension [67] domain of Mcm2 (comprising the histone-binding domain and a linker region, residues 60–150), temporarily relocating detached histones to the front of the replisome before entering the transfer pathway. Disruption of the interaction between Tof1 and the NTE domain of Mcm2 results in an apparent leading strand bias for parH3:H4tet deposition, indicating a defect in parental histone transfer to the lagging strand, as observed in eSPAN analysis [9]. Similar to the FACT components, deletion of Tof1 also does not affect the parental histone transfer strand bias in budding yeast, supporting Tof1’s role in both leading and lagging strand parental transfer pathways.

Asf1a, a highly conserved histone chaperone, acts as a chromatin modifier hub for a network of chromatin-associated proteins involved in DNA repair, new synthetic histone H3-H4 transfer, nucleosome assembly, and disassembly [68]. Mcm2 also binds with Asf1a [69], bridging by histone H3-H4 [70], suggesting a potential role for Asf1a in parental histone transferring [71]. However, direct evidence for Asf1a’s involvement in parental histone transferring is lacking. Generally, Asf1a is believed to be involved in new H3-H4 deposition and modification [72]. Further studies will be needed to clarify whether Asf1 is involved in parental H3-H4 transfer.

## 6. Consequence of Defects in Parental Histone Transfer

Parental histone transfer is generally considered crucial for epigenetic inheritance. However, defects in parental histone transfer mutants, such as mcm2-3A or deletions of Dpb3 or Dpb4, only marginally increase the loss of silence at mating-type locus or telomere regions. Furthermore, the mcm2-3A and dpb3Δ double mutant with a balanced parental histone transfer pattern exhibits increased loss of silencing compared to single mutants [52,73]. Minor phenotypic effects of parental histone transfer mutants in budding yeast are expected, as the parental histone transfer pathway would have a profound impact on DNA sequence-independent inheritance; however, this phenomenon has not yet been detected in budding yeast. In addition to these sequence-dependent gene silencing mechanisms, a histone tail “copy and paste” mechanism has been proposed to explain epigenetic inheritance [73,74]. In mouse cells, the Polycomb repressive complex 2, a major gene silencer, can bind to the histone tail modification histone H3 lysine 27 trimethylation (H3K27me3) and promote further local H3K27me3 modification [75]. Similarly, in the absence of a demethylase, histone lysine H3K9 methyltransferase Clr4 can perform a reader–writer function to copy histone H3 lysine 9 trimethylation (H3K9me3) modifications in fission yeast. 

In the fission yeast heterochromatin inheritance system, mutations of the histone-binding domain of Mcm2 severely disrupt heterochromatin inheritance, whereas yeast cells with mutations in Dpb3/4, proteins that transfer parental histones to the leading strand, exhibit efficient heterochromatin inheritance, indicating that Mcm2 has a stronger influence on epigenetic inheritance compared to Dpb3/4 [76]. As the “copy and paste” mechanism predicts, the mcm2-2A and dpb4Δ double mutant, which has a balanced parental histone transfer pattern, shows much lower loss of silence frequency compared with the single mcm2-2A mutant at the heterochromatic mating-type locus [76]. This observation underscores the importance of the symmetric distribution of parH3:H4tet and the density of H3-H4 histones at daughter strands for epigenetic inheritance. Heterochromatin instability in higher organisms, including fission yeast, involves complex interactions across multiple gene silencing pathways, such as the “copy and paste” mechanism. In fission yeast, mutations in Mcm2 and Dpb3/4 have only minor effects on pericentric heterochromatin integrity, which differs from their impact on the mating type locus. This difference is likely due to the overlapping function of the RNAi pathway in heterochromatin initiation at the pericentric loci [76]. Nevertheless, this fission yeast study also suggests that parental histone transfer pathways play an important role in regulating chromatin structure and gene expression (Figure 3).

In budding yeast cells, mutations in parental histone transfer pathways do not significantly alter global nucleosome positioning and occupancy compared with wild-type cells. Additionally, only slight increases in spontaneous DNA strand breaks were observed in cells with mcm2-2A or dpb3 Δ mutations, and there was no systemic DNA damage response in these mutated cells [52]. However, they had a significantly decreased frequency of homologous recombination. This can be explained by the elevated free histone levels resulting from disrupted parental histone deposition pathways, which in turn suppress frequency of homologous recombination [64]. These results highlight a connection between the parental histone transfer process and the regulation of DNA repair pathways [52].

The parental histone transfer process is expected to impact epigenomic stability, potentially affecting cellular differentiation. Several studies have investigated the influence of the parental nucleosome assembly pathway on cellular differentiation and development. Both mcm2-2A and Pol E3 (homologs of yeast Dpb4) deletion ES cells have shown reduced plasticity [58,77,78] (Figure 3). The asymmetric parental distribution on ES cell differentiation likely involves changes in the occupancy of post-translational modification (PTM) enzymes, which contribute to alterations in PTM distribution in newly synthesized nucleosomes. Since changes in active modifications can lead to differential gene expression, such as H3K27me3 regulating bivalent genes and H3K9me3 regulating repeat repression, asymmetric parental histone distribution affects the active and repressive modifications at bivalent promoters, resulting in global changes in gene expression [58]. Mcm2 is located at the transcription start sites of actively transcribed regions, and mutation of the histone-binding domain significantly reduces this localization due to reduced chromatin accessibility [77]. This variation is also related to asymmetric parental histone transfer affected by mutations in the histone-binding domain [58,78].

Furthermore, heterozygous Mcm2-2A mutant mice have been generated. As expected, homozygous Mcm2-2A mutant mice exhibit early embryonic developmental defects and embryonic lethality [58] (Figure 3). Based on these findings, it was concluded that the symmetric distribution and inheritance of parental histones play a pivotal role in early embryo development. Given the fundamental role of cellular differentiation and development in the survival of multicellular organisms, it is reasonable to anticipate that nucleosome assembly plays a widespread role in differentiation across various phyla. Asymmetric cell division is sometimes necessary to establish and maintain different gene expression programs in various cell lineages during the development of multicellular organisms. Disruption of the asymmetric distribution of parental histones in Drosophila germline stem cells, for example, has been associated with the onset of early germline tumors and the loss of germline stem cells [67]. These findings suggest that histone modification bias resulting from asymmetry in histone allocation may have significant biological implications.

## 7. Conclusions

The MCM2-7 helicase subunit Mcm2 plays multiple roles in cellular functions beyond its fundamental helicase activity. One of its crucial roles is mediating the recycling of parH3:H4tet, a process highly conserved from yeast to human cells. Multiple factors that interact with the MCM2-7 helicase are also involved in parental histone transfer. The physiological significance of this parental histone transfer includes, but is not limited to, maintaining epigenome stability, influencing choices of DNA repair pathways, and regulating cellular differentiation and embryonic development. Based on the current literature, it is suggested that the replisome is not only essential for stable inheritance of genetic information, but also for epigenetic inheritance. Until now, all known regulators of parH3:H4tet transfer are replisome components, including Mcm2. Whether other metabolic pathways are also involved in this regulation remains to be explored. Additionally, it should be clarified whether the parental H3-H4 and H2A-H2B can be co-transferred during DNA replication, especially in light of the discovery of the hexamer structure of (H3-H4)2-(H2A-H2B) [9]. Another important issue is to determine the effect of transcription–replication conflicts on chromatin replication [79]. It is essential to investigate whether RNA transcription affects chromatin replication, including the recycling of parH3:H4tet.

## Figures and Tables

**Figure 1 biology-13-00572-f001:**
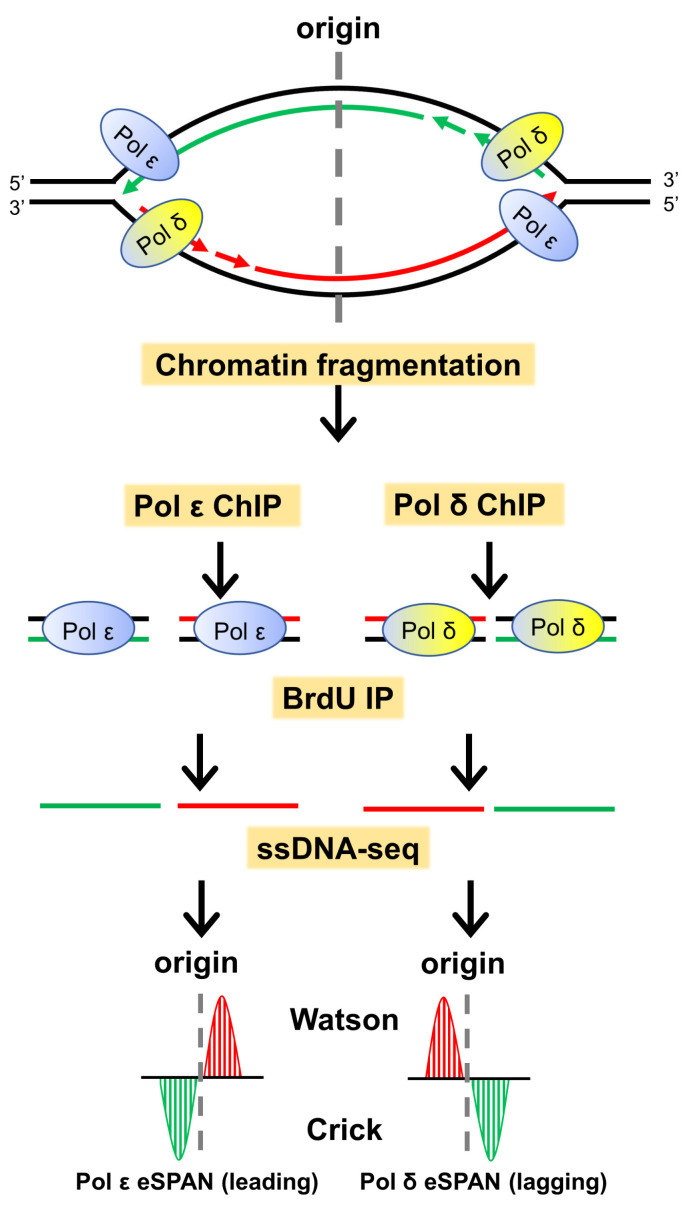
Outline of the experimental strategy for the eSPAN method, with leading and lagging strand polymerases (Pol δ and Pol ε) as examples. During cell culture, the newly synthesized DNA is labeled with bromodeoxyuridine (BrdU). The replicating chromatin is fragmented with physical sonication or micrococcal nuclease (MNase) digestion. The Pol δ and Pol ε ChIP process is the same as for standard chromatin immunoprecipitation (ChIP)-seq. After DNA extraction of protein ChIP, the DNA is denatured, and newly synthesized DNA is recovered with BrdU immunoprecipitation (IP) using a BrdU-specific antibody. The processed short-stranded DNA (ssDNA) is constructed into an ssDNA library, in which the strand information is kept. After sequencing, the reads are mapped to the Watson and Crick strands of the budding yeast genome to determine the location of the target protein and strand-specific information. Therefore, eSPAN detects the association of a protein with nascent DNA at DNA replication forks. In this cartoon, the red line represents a new Watson strand; the green line represents a new Crick strand; and the black line represents parental DNA.

**Figure 2 biology-13-00572-f002:**
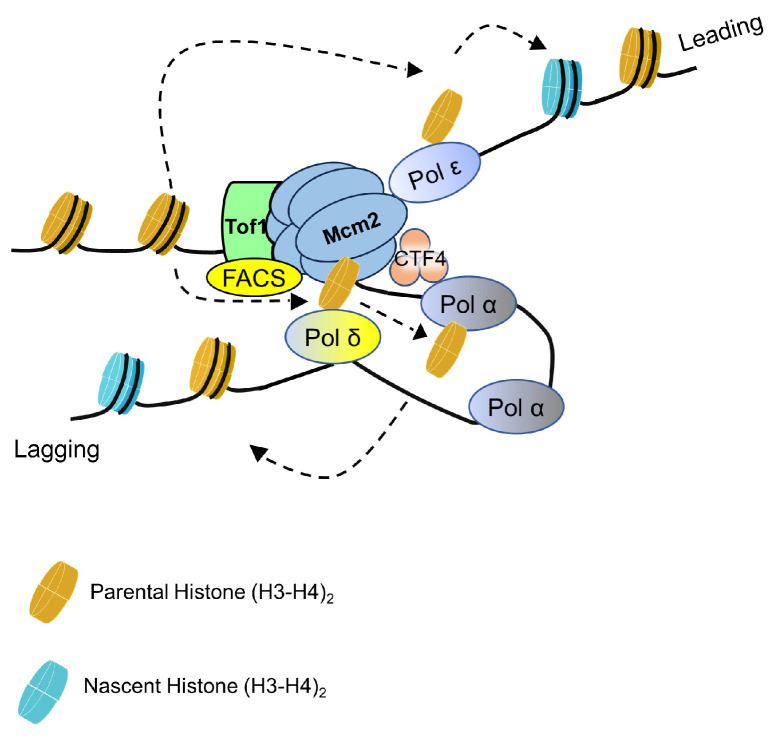
A cartoon of the parental histone transfer pathway and parental chaperones.

**Figure 3 biology-13-00572-f003:**
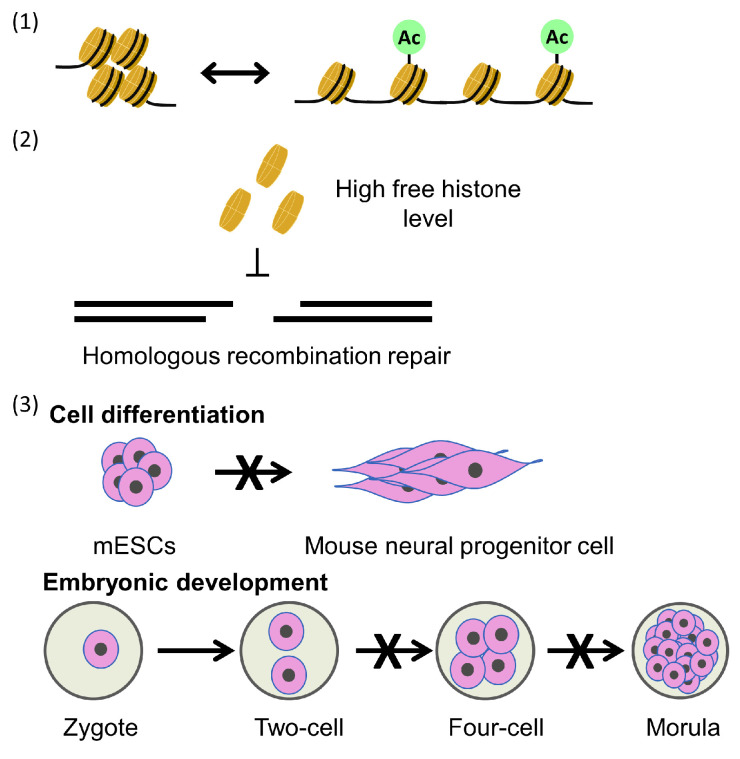
Consequences of parental histone defects: (1) changes in chromatin structure and gene expression patterns; (2) defects of parental histone transfer leading to high levels of cell-free histones, which can inhibit homologous recombination; and (3) defects of parental histone transfer can lead to defects in mESC differentiation and development of mouse embryonic defects.

**Table 1 biology-13-00572-t001:** List of gene mutations.

Gene	Deletion/Mutation	Mutation	Phenotype	Other Organisms
CTF4	ctf4-4E	Point mutations: L867E, A871E, A897E and I901E	CTF4-CMG helicase and CTF4-Pol1p interactions are disrupted. parH3:H4tet show a leading strand bias	
Dpb3	dpb3∆		parH3:H4tet show a lagging strand bias	mouse
Dpb4	dpb4∆		parH3:H4tet show a lagging strand bias	mouse
Spt16	Spt16 N-terminal deletion	Deletion of Spt16 N-terminal region	a slight a leading strand bias	
Mcm2	mcm2-3A	Point mutations Y79A, Y82A, and Y91A	parH3:H4tet show a leading strand bias	Y81A, Y89A, and Y137A in humans, mouse
	mcm2-2A	Point mutations Y79A, Y82A	parH3:H4tet show a leading strand bias	Y81A, Y89A in humans, mouse
Pol1	pol1-4A	Point mutations D141A, D142A, L144A and F147A	CTF4-Pol α interaction is disrupted. parH3:H4tet show a leading strand bias	

## Data Availability

Not applicable.

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
