# Peer review of "The Role of the MCM2-7 Helicase Subunit MCM2 in Epigenetic Inheritance"

_biology, 2024, doi:10.3390/biology13080572_

Round 1
Reviewer 1 Report
Comments and Suggestions for Authors
Jia and Yu have written a comprehensive review on the role of the eukaryotic DNA replisome in epigenetic inheritance. The manuscript highlights the fundamental role of the eSPAN technique in distinguishing between proteins that associate with the leading strand and those that associate with the lagging strand during DNA replication. It addresses the central role of the Mcm2 subunit of the CMG helicase as a chaperone element and describes the differential interactors and consequences of mutations. The review is well written and up to date. I have only a few suggestions to improve the readability and interest of the text.
Since the main focus of the review is the role of Mcm2, I think it will be beneficial to include this fact in the title, which seems to be very general as it is.
Page 1 line 33 ‘although recent reports question this point’ It might be informative for the reader to see how the point is questioned and how it is resolved by others.
Page 2 line 64. I think the authors could include a short paragraph at the end of the introduction to actually introduce that the review is about the role of Mcm2. This will help the reader to focus on the rest of the text.
Page 3 line104, line 106, line 108 and others in the text. There is a lack of consistency in how the authors refer to the helicase core. Mcm2-7, MCM2-7, and MCM are used throughout the text.
Page 4 line 154. This sentence requires references
Page 5 line 208. Hexameric complex?
Page 6 line 217. Delete the word ‘am’.
Author Response
Since the main focus of the review is the role of Mcm2, I think it will be beneficial to include this fact in the title, which seems to be very general as it is.
Response: Thank you for your advice, I have revised the title: The Role of the MCM Helicase Subunit Mcm2 in Epigenetic Inheritance.
Page 1 line 33 ‘although recent reports question this point’ It might be informative for the reader to see how the point is questioned and how it is resolved by others.
Response: Thank you for the suggestion. The relevant words have been added as “In the recent studies, the histone H2A-H2B was found to segregated symmetrically to daughter strands. Furthermore, the structure analysis implied that the parental H2A-H2B dimers could be recycled either as a form of hexamer with parental H3-H4 tetramer [8,9].”
Page 2 line 64. I think the authors could include a short paragraph at the end of the introduction to actually introduce that the review is about the role of Mcm2. This will help the reader to focus on the rest of the text.
Response: Thank you for the suggestion. A short description has been added as “In recent years, advances in technology have enabled the exploration of mechanisms underlying the recycling of parental histone H3-H4. In this manuscript, we will focus on the role of Mcm2 in this process.”
Page 3 line104, line 106, line 108 and others in the text. There is a lack of consistency in how the authors refer to the helicase core. Mcm2-7, MCM2-7, and MCM are used throughout the text.
Response: Thank you for your suggestion, the name of MCM2-7 has been used in place of other names.
Page 4 line 154. This sentence requires references
Response: We have added the references.
Page 5 line 208. Hexameric complex?
Response: We made the edition as suggested.
Page 6 line 217. Delete the word ‘am’.
Response: We made the edition as suggested.
Reviewer 2 Report
Comments and Suggestions for Authors
This comprehensive review synthesizes recent findings regarding the pivotal role of the replication machinery, with particular emphasis on the MCM helicase, in epigenetic inheritance. The article discusses how these molecular components contribute to the maintenance and transmission of epigenetic information across generations. Additionally, the review explores emerging research directions and potential future avenues in this field, highlighting key questions and opportunities for further investigation. By integrating both current insights and prospective considerations, this review aims to provide a holistic understanding of the dynamic interplay between replication machinery and epigenetic phenomena, paving the way for future advancements in our knowledge and applications.
This version summarizes the current understanding but I would recommend to add a forward-looking perspective on future research directions, enhancing the depth and relevance of the review article.
Comments on the Quality of English LanguageIt is fine.
Author Response
This version summarizes the current understanding but I would recommend to add a forward-looking perspective on future research directions, enhancing the depth and relevance of the review article.
Response: Thank you for your suggestion. I have a short perspective on the future directions in the conclusion parts:”Until now, all known regulators of parental H3-H4 transfer are replisome components, including Mcm2. Whether other metabolic pathways are also involved in this regulation remains to be explored. Additionally, it should be clarified whether the parental H3-H4 and H2A-H2B can be co-transferred during DNA replication, especially in light of the discovery of the hexamer structure of (H3-H4)2-(H2A-H2B). Another important issue is the transcription-replication conflict. It is essential to investigate whether RNA transcription affects chromatin replication, including the recycling of parental H3-H4”.
Reviewer 3 Report
Comments and Suggestions for Authors
Recycling parental histones is a crucial step in chromosome replication and is essential for epigenetic inheritance. Parental histones H2A/H2B and H3/H4 are recycled through different mechanisms. In this review, Jia and Yu focus on the recycling of the parental histone H3-H4 tetramer and summarize recent knowledge of the recycling pathway for both the leading and lagging strands at the replication fork. They highlight the role of MCM helicase subunit Mcm2 and its interaction partners in epigenetic inheritance. Disruption of these histone chaperones' functions leads to changes in chromatin structure and gene expression patterns, higher levels of cell-free histones, and defects in cellular differentiation and embryonic development. This underscores the physiological significance of these parental histone transfer pathways in maintaining epigenomic stability. The review is well-organized and provides essential background knowledge of eSPAN, a method used for the genome-wide analysis of replication-coupled distribution of parental histones on leading and lagging strands. Recently published papers on related topics are referenced and discussed. This review is ready for publication in its current form and will provide the field with an up-to-date resource.
Author Response
Response: Thank you for your recommendations.